# The protein domains of vertebrate species in which selection is more effective have greater intrinsic structural disorder

Catherine A Weibel[1,2†‡], Andrew L Wheeler[3†], Jennifer E James[4§], Sara M Willis[4#], Hanon McShea[5], Joanna Masel[4]*

[1]Department of Mathematics, University of Arizona, Tucson, United States; [2]Department of Physics, University of Arizona, Tucson, United States; [3]Genetics Graduate Interdisciplinary Program, University of Arizona, Tucson, United States; [4]Department of Ecology and Evolutionary Biology, University of Arizona, Tucson, United States; [5]Department of Earth System Science, Stanford University, Stanford, United States

*For correspondence:
masel@arizona.edu

[†]These authors contributed equally to this work

Present address: [‡]Department of Applied Physics, Stanford University, Stanford, United States; [§]Department of Ecology and Genetics, Evolutionary Biology Center, Uppsala University, Uppsala, Sweden; [#]University Information Technology Services, University of Arizona, Tucson, United States

Competing interest: The authors declare that no competing interests exist.

**Abstract** The nearly neutral theory of molecular evolution posits variation among species in the effectiveness of selection. In an idealized model, the census population size determines both this minimum magnitude of the selection coefficient required for deleterious variants to be reliably purged, and the amount of neutral diversity. Empirically, an 'effective population size' is often estimated from the amount of putatively neutral genetic diversity and is assumed to also capture a species' effectiveness of selection. A potentially more direct measure of the effectiveness of selection is the degree to which selection maintains preferred codons. However, past metrics that compare codon bias across species are confounded by among-species variation in %GC content and/or amino acid composition. Here, we propose a new Codon Adaptation Index of Species (CAIS), based on Kullback–Leibler divergence, that corrects for both confounders. We demonstrate the use of CAIS correlations, as well as the Effective Number of Codons, to show that the protein domains of more highly adapted vertebrate species evolve higher intrinsic structural disorder.

## eLife assessment

This study develops a **useful** metric for quantifying codon usage adaptation - the Codon Adaptation Index of Species (CAIS). This metric permits direct comparisons of the strength of selection at the molecular level across species. The study is based on **solid** evidence, and the authors identify relationships between CAIS and the presence of disordered protein domains. Other correlations, such as the one between CAIS and body size, are weak and non-significant. In summary, the study introduces an interesting new approach to quantifying codon usage across species, which may be helpful in attempts to measure selection at the molecular level.

**eLife digest** Evolution is the process through which populations change over time, starting with mutations in the genetic sequence of an organism. Many of these mutations harm the survival and reproduction of an organism, but only by a very small amount.

Some species, especially those with large populations, can purge these slightly harmful mutations more effectively than other species. This fact has been used by the 'drift barrier theory' to explain various profound differences amongst species, including differences in biological complexity. In this theory, the effectiveness of eliminating slightly harmful mutations is specified by an 'effective' population size, which depends on factors beyond just the number of individuals in the population.

Effective population size is normally calculated from the amount of time a 'neutral' mutation (one with no effect at all) stays in the population before becoming lost or taking over. Estimating this time requires both representative data for genetic diversity and knowledge of the mutation rate. A major limitation is that these data are unavailable for most species. A second limitation is that a brief, temporary reduction in the number of individuals has an oversized impact on the metric, relative to its impact on the number of slighly harmful mutations accumulated.

Weibel, Wheeler et al. developed a new metric to more directly determine how effectively a species purges slightly harmful mutations. Their approach is based on the fact that the genetic code has 'synonymous' sequences. These sequences code for the same amino acid building block, with one of these sequences being only slightly preferred over others.

The metric by Weibel, Wheeler et al. quantifies the proportion of the genome from which less preferred synonymous sequences have been effectively purged. It judges a population to have a higher effective population size when the usage of synonymous sequences departs further from the usage predicted from mutational processes.

The researchers expected that natural selection would favour 'ordered' proteins with robust three-dimensional structures, i.e., that species with a higher effective population size would tend to have more ordered versions of a protein. Instead, they found the opposite: species with a higher effective population size tend to have more disordered versions of the same protein. This changes our view of how natural selection acts on proteins.

Why species are so different remains a fundamental question in biology. Weibel, Wheeler et al. provide a useful tool for future applications of drift barrier theory to a broad range of ways that species differ.

## Introduction

Species differ from each other in many ways, including mating system, ploidy, spatial distribution, life history, size, lifespan, genome size, mutation rate, selective pressure, and population size. These differences make the process of purifying selection more efficient in some species than others. Our understanding of both the causes and consequences of these differences is limited in part by a reliable metric with which to measure them. In the long term, the probability that a gene is fixed for one allele rather than another allele is given by the ratio of fixation and counter-fixation probabilities (*Bulmer, 1991*). In an idealized population of constant population size and no selection at linked sites, a mutation–selection–drift model describes how this ratio of fixation probabilities depends on the census population size $N$ (*Kimura, 1962*), and hence gives the fraction of sites expected to be found in preferred vs. non-preferred states (*Figure 1*).

This reasoning has been extended to real populations by positing that species have an 'effective' population size, $N_e$ (*Ohta, 1973*). $N_e$ is the census population size of an idealized population that reproduces a property of interest in the focal population. $N_e$ is therefore not a single quantity per population, but instead depends on which property is of interest.

The amount of neutral polymorphism is the usual property used to empirically estimate $N_e$ (*Charlesworth, 2009*; *Doyle et al., 2015*; *Lynch et al., 2016*). However, the property of most relevance to nearly neutral theory is instead the inflection point $s$ at which non-preferred alleles become common enough to matter (*Figure 1*), and hence the degree to which highly exquisite adaptation can be maintained in the face of ongoing mutation and genetic drift (*Kimura, 1962*; *Ohta, 1972*; *Ohta, 1992*). While genetic diversity has been found to reflect some aspects of life history strategy

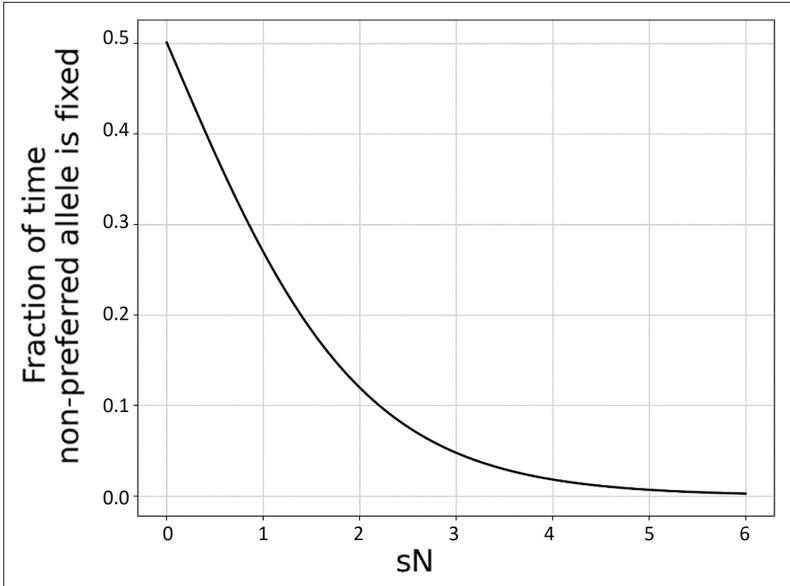

**Figure 1.** The effectiveness of selection, calculated as the long-term ratio of time spent in fixed deleterious: fixed beneficial allele states given symmetric mutation rates, is a function of the product $sN$. Assuming a diploid Wright–Fisher population with $s << 1$, the probability of fixation of a new mutation $\pi\left(N, s\right) = \frac{1-e^{-\frac{s}{2}}}{1-e^{-Ns}}$, and the $y$-axis is calculated as $\pi\left(N, -s\right) / \left(\pi\left(N, -s\right) + \pi\left(N, s\right)\right)$. $s$ is held constant at a value of 0.001 and $N$ is varied. Results for other small magnitude values of $s$ are superimposable. For small $sN$, selection is ineffective at producing codon bias. For large $sN$, selection is highly effective. For only a relatively narrow range of intermediate values of $sN$, the degree of codon bias depends quantitatively on $sN$.

(*Romiguier et al., 2014*), there remain concerns about whether neutral genetic diversity and the limits to weak selection always remain closely coupled in non-equilibrium settings.

As a practical matter, $N_e$ is usually calculated by dividing some measure of the amount of putatively neutral (often synonymous) polymorphism segregating in a population by that species' mutation rate (*Charlesworth, 2009*). As a result, $N_e$ values are only available for species that have both polymorphism data and accurate mutation rate estimates, limiting their use. Worse, $N_e$ is not a robust statistic. In the absence of a clear species definition, polymorphism is sometimes calculated across too broad a range of genomes, substantially inflating $N_e$ (*Daubin and Moran, 2004*); a poor sampling scheme can have the converse effect of deflating genetic diversity. Transient hypermutation (*Plotkin et al., 2006*), which is common in microbes, causes further short-term inconsistencies in polymorphism levels. Perhaps most importantly, a recent bottleneck will deflate $N_e$ based on the coalescence time, even if too brief to lead to significant erosion of fine-tuned adaptations. But drift barrier theory concerns the level with which adaptation is fine-tuned, and so a better metric would capture that directly, rather than indirectly rely on neutral diversity.

An alternative approach to measure the efficiency of selection exploits codon usage bias, which is influenced by weak selection for factors such as translational speed and accuracy (*Hershberg and Petrov, 2008*; *Plotkin and Kudla, 2011*; *Hunt et al., 2014*). The degree of bias in synonymous codon usage that is driven by selective preference offers a more direct way to assess how effective selection is at the molecular level in a given species (*Li, 1987*; *Bulmer, 1991*; *Akashi, 1996*; *Subramanian, 2008*). Conveniently, it can be estimated from only a single genome, that is, without polymorphism or mutation rate data for that species.

One commonly used metric, the Codon Adaptation Index (CAI) (*Sharp and Li, 1987*; *Sharp et al., 2010*) takes the average of Relative Synonymous Codon Usage (RSCU) scores, which quantify how often a codon is used, relative to the codon that is most frequently used to encode that amino acid in that species. While this works well for comparing genes within the same species, it unfortunately means that the species-wide strength of codon bias appears in the normalizing denominator (see *Equation 4* and *Figure 3—figure supplement 1A*). Paradoxically, this can make more exquisitely adapted species

have lower rather than higher species-averaged CAI scores (*Figure 3—figure supplement 1B*; *Rocha, 2004*; *Botzman and Margalit, 2011*).

To compare species using CAI, it has been suggested that instead of taking a genome-wide average, one should consider a set of highly expressed reference genes (*Sharp et al., 2005*; *Vicario et al., 2007*; *Subramanian, 2008*; *dos Reis and Wernisch, 2009*). This approach assumes that the relative strength of selection on those reference genes (often a function of gene expression) remains approximately constant across the set of species considered (red distributions in *Figure 2*). Its use also requires careful attention to the length of reference genes (*Urrutia and Hurst, 2001*; *Doherty and McInerney, 2013*), and some approaches also require information about tRNA gene copy numbers and abundances (*dos Reis and Wernisch, 2009*).

Since codon bias varies quantitatively within only a small range of $sN$ (*Figure 1*), a promising approach is to measure the proportion of sites at which codon adaptation is effective. We posit that more highly adapted species have a higher proportion of both genes and sites subject to effective selection on codon bias (*Figure 2*; *Galtier et al., 2018*). Indeed, CAI might also rely in part on variation in the fraction of sites within the reference genes that is subject to effective selection as a function of species (*Figure 2*, red). Here we take this logic further, considering all sites in a proteome-wide approach. Averaging across the entire proteome provides robustness to shifts in the expression level of or strength of selection on particular genes. The proteome-wide average depends on the fraction of sites whose selection coefficients exceed the 'drift barrier' for that particular species (*Figure 2*, blue threshold).

In estimating the effects of selection, it is critical to control for other causes of codon bias. In particular, species differ in their mutational bias with respect to the proportion of the genome that consists of guanine-cytosine base pairs (GC), and in the frequency of GC-biased gene conversion (*Urrutia and Hurst, 2001*; *Duret and Galtier, 2009*; *Doherty and McInerney, 2013*; *Figuet et al., 2014*). Here, we control for %GC, capturing species differences both in mutation and in gene conversion, by calculating the Kullback–Leibler divergence of the observed codon frequencies away from the codon

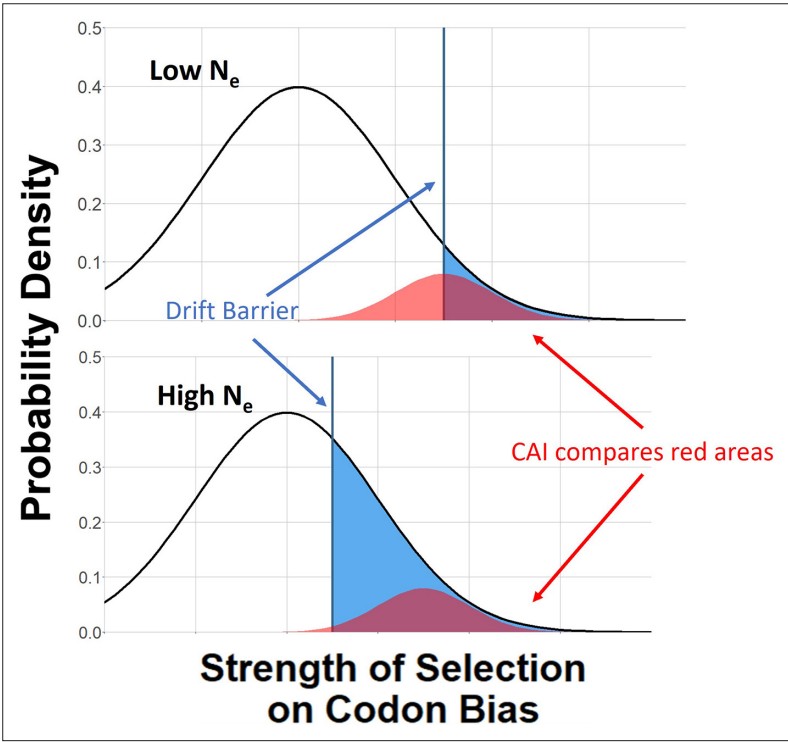

**Figure 2.** More highly adapted species (bottom) have a higher proportion of their sites subject to effective selection on codon bias (blue area). The Codon Adaptation Index (CAI) attempts to compare the intensity of selection (*Figure 1*, x-axis) in a subset of genes under strong selection (red areas). Given the narrow range of quantitative dependence of codon bias on $sN$ shown in *Figure 1*, our new metric is intended to capture differences in the proportion of the proteome subject to substantial selection (blue areas).

frequencies that we would expect to see given the genomic %GC content of the species. Kullback–Leibler divergence measures the distance of an observed probability distribution from an expected reference distribution, capturing a measure of surprise (*Kullback and Leibler, 1951*). This method does not require us to specify preferred vs. non-preferred codons, and can thus also accommodate situations in which different genes have different codon preferences (*Gingold et al., 2014*; *Cope et al., 2018*).

An alternative metric, the Effective Number of Codons (ENC) originally quantified how far the codon usage of a sequence departs from equal usage of synonymous codons (*Wright, 1990*), with lower ENC values indicating greater departure. This approach creates a complex relationship with GC content (*Fuglsang, 2008*), and so ENC was later modified to correct for GC content (*Novembre, 2002*). However, a remaining issue with this modified ENC is that differences among species in amino acid composition might act as a confounding factor, even after controlling for GC content. Specifically, species that make more use of an amino acid for which there is stronger selection among codons (which is sometimes the case *Vicario et al., 2007*) would have higher codon bias, even if each amino acid considered on its own had identical codon bias irrespective of which species it is in. Confounding with amino acid frequencies has been shown to be a problem at the individual protein level (*Cope et al., 2018*). Neither ENC (*Fuglsang, 2004*; *Fuglsang, 2008*) nor the CAI (*Sharp and Li, 1987*) adequately control for differences in amino acid composition when applied across species. Despite early claims to the contrary (*Wright, 1990*), this problem is not easy to fix for ENC (*Fuglsang, 2004*; *Fuglsang, 2008*).

Here, we extend the CAI, using the information-theory-based Kullback–Leibler divergence, so that it corrects for both GC and amino acid composition (see Methods) to create a new Codon Adaptation Index of Species (CAIS). The availability of a complete genome allows both metrics to be readily calculated without data on polymorphism or mutation rate, without selecting reference genes, and without concerns about demographic history. Our purpose is to find an accessible metric that can quantify the limits to weak selection important to nearly neutral theory; this differs from past evaluations focused

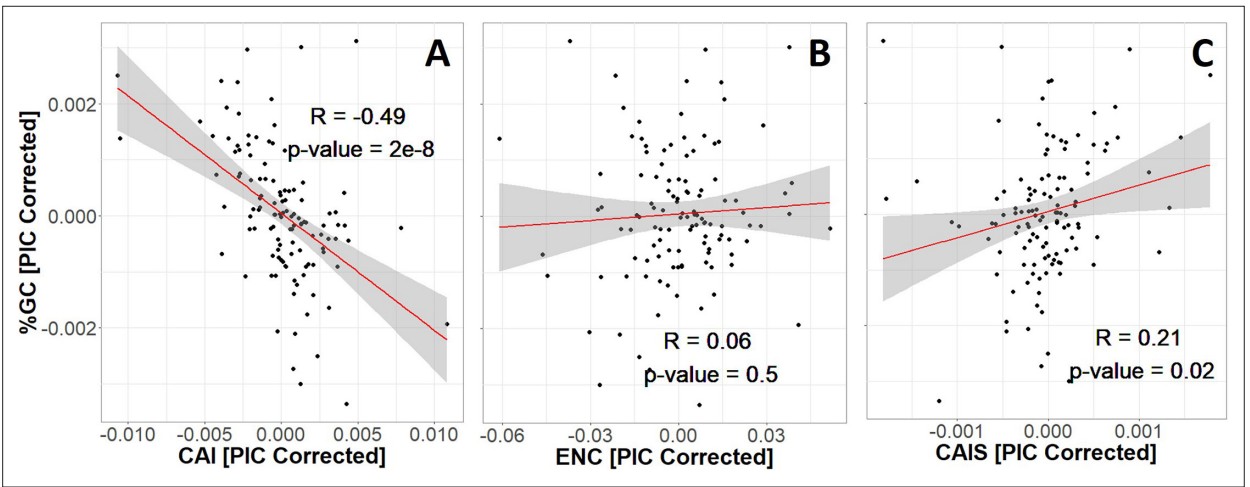

**Figure 3.** Codon Adaptation Index (CAI) is seriously confounded with GC content (**A**), while Effective Number of Codons (ENC) and Codon Adaptation Index of Species (CAIS) are not (**B** and **C**). We control for phylogenetic confounding via Phylogenetic Independent Contrasts (PIC) (*Felsenstein, 1985*); this yields an unbiased $R^2$ estimate (*Rohlf, 2006*). Each datapoint is one of 118 vertebrate species with 'Complete' intergenic genomic sequence (allowing for %GC correction) and TimeTree divergence dates (allowing for PIC correction). Red line shows unweighted $lm(y \sim x)$ with gray region as 95% confidence interval. *Figure 3—figure supplement 1* shows in more detail why CAI is not appropriate for species-wide effectiveness of selection measurements. Plots without PIC correction are shown in *Figure 3—figure supplement 2*. The impact of amino acid frequency correction on CAIS is shown in *Figure 3—figure supplement 3*.

The online version of this article includes the following figure supplement(s) for figure 3:

**Figure supplement 1.** Codon Adaptation Index (CAI) is not appropriate for species-wide effectiveness of selection measurements.

**Figure supplement 2.** The same relationships are shown as in *Figure 3*, but without correction for phylogenetic confounding, suggesting GC confounding for the Effective Number of Codons (ENC) but not the Codon Adaptation Index of Species (CAIS).

**Figure supplement 3.** Vertebrate Codon Adaptation Index of Species (CAIS) values are not greatly affected by computation for a standardized amino acid composition vs. computation for the amino acid frequencies in the species in question.

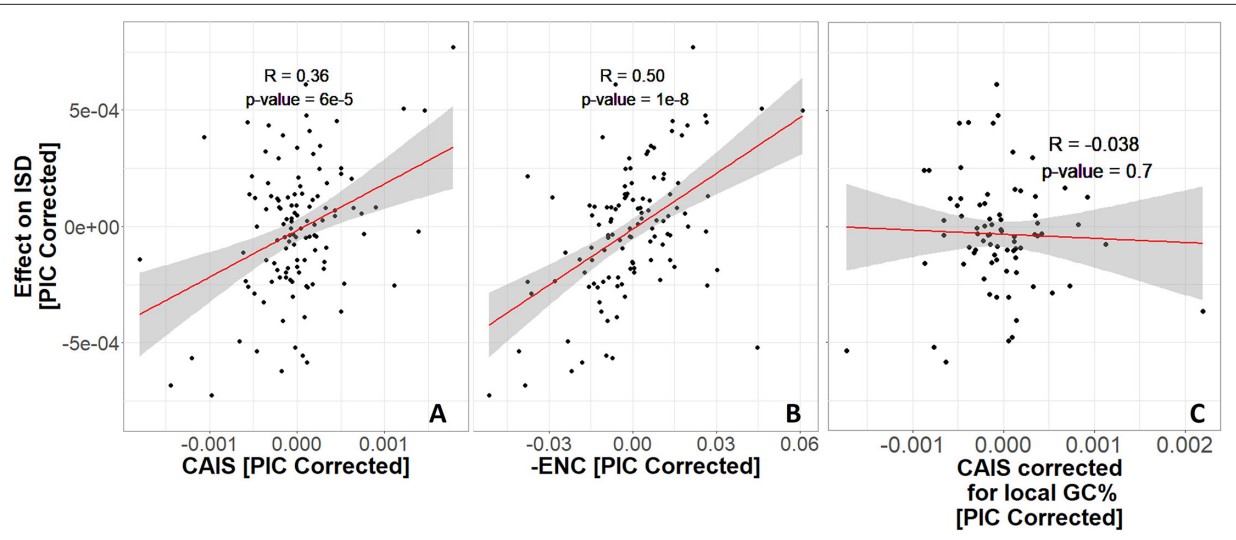

**Figure 4.** Protein domains have higher intrinsic structural disorder (ISD) when found in more exquisitely adapted species, according to (**A**) the Codon Adaptation Index of Species (CAIS) and (**B**) the Effective Number of Codons (ENC). We plot -ENC rather than ENC to more easily compare results with those from CAIS. (**C**) Correcting for local rather than genome-wide %GC removes the relationship. Each datapoint is one of 118 vertebrate species with 'complete' intergenic genomic sequence available (allowing for %GC correction), and TimeTree divergence dates (allowing for Phylogenetic Independent Contrasts [PIC] correction). 'Effects' on ISD shown on the *y*-axis are fixed effects of species identity in our linear mixed model, after PIC correction. Red line shows unweighted lm(*y* ~ *x*) with gray region as 95% confidence interval. Panels without PIC correction are presented in *Figure 4—figure supplement 1*.

The online version of this article includes the following figure supplement(s) for figure 4:

**Figure supplement 1.** The same relationships are shown as in *Figure 4*, here without correction for phylogenetic confounding.

on comparing different genes of the same species and recapitulating 'ground truth' simulations thereof (*Sun et al., 2013*; *Zhang et al., 2012*; *Liu et al., 2018*). To demonstrate the usefulness of our method, we identify a novel correlation with intrinsic structural disorder (ISD), pointing to what else might be subject to weak selective preferences at the molecular level. While ENC can also identify subtle selection on ISD, CAIS can do so without the risk of confounding with amino acid frequencies.

## Results

### Both ENC and CAIS solve the GC confounding problem that plagues CAI

CAI is seriously confounded with GC content (*Figure 3A*). ENC is not confounded with GC content (*Figure 3B*), while CAIS has only a very weak correlation that is not significant after correction for multiple comparisons (*Figure 3C*).

### Proteins in better adapted species evolve more structural disorder

As an example of how correlations with codon adaptation metrics can be used to identify weak selective preferences, we investigate protein ISD. Disordered proteins are more likely to be harmful when overexpressed (*Vavouri et al., 2009*), and ISD is more abundant in eukaryotic than prokaryotic proteins (*Schad et al., 2011*; *Xue et al., 2012*; *Basile et al., 2019*), suggesting that low ISD might be favored by more effective selection.

However, compositional differences among proteomes might not be driven by differences in how a given protein sequence evolves as a function of the effectiveness of selection. Instead, they might be driven by the recent birth of ISD-rich proteins in animals (*James et al., 2021*), and/or by differences among sequences in their subsequent tendency to proliferate into many different genes (*James et al., 2023*). To focus only on the effects of descent with modification, we use a linear mixed model, with each species having a fixed effect on ISD, while controlling for Pfam domain identity as a random effect. We note that once GC is controlled for, codon adaptation can be assessed similarly

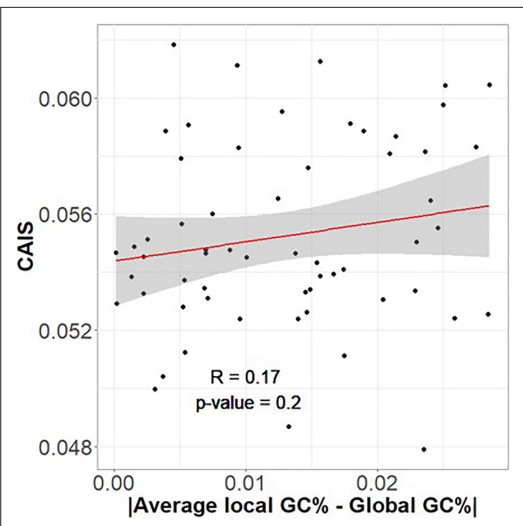

**Figure 5.** Codon Adaptation Index of Species (CAIS) is not correlated with the degree to which local genomic regions differ in their GC content from global GC content. If CAIS were driven by GC-biased gene conversion, genomes with more heterogeneous %GC distributions should have higher CAIS scores.

in intrinsically disordered vs. ordered proteins (*Gossmann et al., 2012*). Controlling for Pfam identity is supported, with standard deviation in ISD of 0.178 among Pfams compared to residual standard deviation of 0.058, and a p-value on the significance of the Pfam random effect term of $3 \times 10^{-13}$. Controlling in this way for Pfam identity, we then ask whether the fixed species effects on ISD are correlated with CAIS and with ENC.

Surprisingly, more exquisitely adapted species have more disordered protein domains (*Figure 4*). Results using ENC and CAIS are similar, with ENC having higher power; the correlation coefficient is 0.36 for CAIS compared to 0.50 for ENC, and the p-value for ENC is 3 orders of magnitude lower. We note, however, that amino acid frequencies strongly influence ISD (*Theillet et al., 2013*). The CAIS correlation is more reliable than the ENC correlation because by construction, CAIS controls for differences in amino acid frequencies among species.

Different parts of the genome have different GC contents (*Bernardi, 2000*; *Eyre-Walker and Hurst, 2001*; *Lander et al., 2001*), primarily because the extent to which GC-biased gene conversion increases GC content depends on the local rate of recombination (*Galtier et al., 2001*; *Meunier and Duret, 2004*; *Duret et al., 2006*; *Duret and Galtier, 2009*). We therefore also calculated a version of CAIS whose codon frequency expectations are based on local intergenic GC content. This performed worse (*Figure 4C*) than our simple use of genome-wide GC content (*Figure 4A*) with respect to the strength of correlation between CAIS and ISD. If GC-biased gene conversion is a more powerful force than weak selective preferences among codons, then local GC content will evolve more rapidly than codon usage (*Kondrashov et al., 2010*). In this case, genome-wide GC may serve as an appropriately time-averaged proxy. It is also possible that the local non-coding sequences we used were too short (at 3000 bp or more), creating excessive noise that obscured the signal.

Many vertebrates have higher recombination rates and hence GC-biased gene conversion near genes; in this case genome-wide GC content would misestimate the codon usage expected from the combination of mutation bias and GC-biased gene conversion in the vicinity of genes. If GC-biased gene conversion drove CAIS, we expect high $\overline{|\text{local GC} - \text{global GC}|}$ to predict high CAIS. We do not see this relationship (*Figure 5*), suggesting that gene conversion strength is not a confounding factor impacting CAIS.

Younger animal-specific protein domains have higher ISD (*James et al., 2021*). It is possible that selection in favor of high ISD is strongest in young domains, which might use more primitive methods to avoid aggregation (*Foy et al., 2019*; *Bertram and Masel, 2020*). To test this, we analyze two subsets of our data: those that emerged prior to the last eukaryotic common ancestor (LECA), here referred to as 'old' protein domains, and 'young' protein domains that emerged after the divergence of animals and fungi from plants. Young and old domains show equally strong trends of increasing disorder with species' adaptedness (*Figure 6*).

## Discussion

When different properties are each causally affected by a species' exquisiteness of adaptation, this will create a correlation between the properties. We use codon adaptation as a reference property, such that correlations with codon adaptation indicate selection. To detect ISD as a novel property under selection, we used a linear mixed model approach that controls for Pfam identity as a random effect.

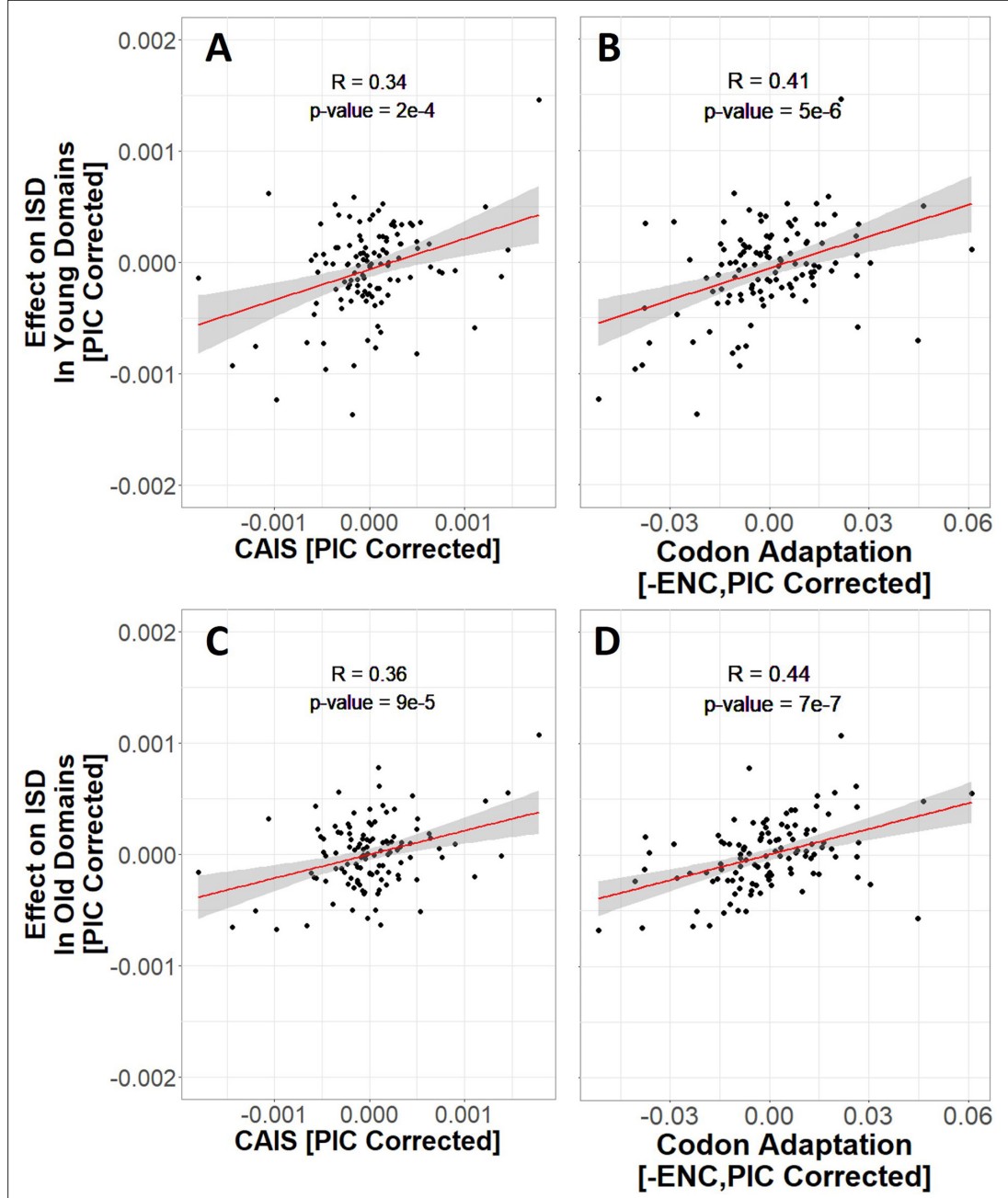

**Figure 6.** More exquisitely adapted species have higher intrinsic structural disorder (ISD) in both young (**A** and **B**) and old (**C** and **D**) protein domains, according to both the Codon Adaptation Index of Species (CAIS) (**A, C**), and the Effective Number of Codons (ENC) (**B, D**). Age assignments are taken from *James et al., 2021*, with vertebrate protein domains that emerged prior to last eukaryotic common ancestor (LECA) classified as 'old', and vertebrate protein domains that emerged after the divergence of animals and fungi from plants as 'young'. 'Effects' on ISD shown on the *y*-axis are fixed effects of species identity in our linear mixed model. The same *n* = 118 datapoints are shown as in *Figures 3 and 4*. Red line shows lm(*y* ~ *x*), with gray region as 95% confidence interval. Panels without Phylogenetic Independent Contrasts (PIC) correction are shown in *Figure 6—figure supplement 1*.

The online version of this article includes the following figure supplement(s) for figure 6:

**Figure supplement 1.** Without correction for phylogenetic confounding, more highly adapted species have higher intrinsic structural disorder (ISD) in both young (**A** and **B**) and old (**C** and **D**) protein domains, according to both the Codon Adaptation Index of Species (CAIS) (**A, C**), and the Effective Number of Codons (ENC) (**B, D**).

This approach shows that the same Pfam domain tends to be more disordered when found in a well-adapted species (i.e. a species with a higher CAIS or ENC). This is true for both ancient and recently emerged protein domains.

It is important that no additional variable such as GC content or amino acid frequencies creates a spurious correlation by affecting both CAIS and our property of interest. For this reason, we define CAIS as the observed Kullback–Leibler divergence (*Kullback and Leibler, 1951*) from the codon usage expected given the GC content. The GC content pertinent to this expectation depends primarily on mutation bias and GC-biased gene conversion (*Romiguier and Roux, 2017*), but potentially also on selection on individual nucleotide substitutions that is hypothesized to favor higher %GC (*Long et al., 2018*). By controlling for %GC, we exclude all these forces from influencing CAIS or ENC. We thus capture the extent of adaptation in codon bias, including translational speed, accuracy, and any intrinsic preference for GC over AT that is specific to coding regions. These remaining codon-adaptive factors do not create a statistically convincing correlation between CAIS and GC (*Figure 3C*), nor between ENC and GC (*Figure 3B*), although CAI is strongly correlated with GC (*Figure 3A*). Notably, our new CAIS metric of codon adaptation controls for amino acid frequencies, rather than, like ENC, only GC content.

A direct effect of ISD on fitness agrees with studies of random Open Reading Frames (ORFs) in *Escherichia coli*, where fitness was driven more by amino acid composition than %GC content, after controlling for the intrinsic correlation between the two (*Kosinski et al., 2022*). However, we have not ruled out a role for selection for higher %GC in ways that are general rather than restricted to coding regions, whether in shaping mutational biases (*Smith and Eyre-Walker, 2001*; *Hershberg and Petrov, 2009*; *Hildebrand et al., 2010*; *Novoa et al., 2019*; *Forcelloni and Giansanti, 2020*) or the extent of gene conversion, or even at the single-nucleotide level in a manner shared between coding regions and intergenic regions (*Long et al., 2018*).

A more complex metric could control for more than just GC content and amino acid frequencies. First vs. second vs. third codon positions have different nucleotide usage on average, but while correcting for this might be useful for comparing genes (*Zhang et al., 2012*), correcting for it while comparing species might remove the effect of interest. Similarly, while it might be useful to control for dinucleotide and trinucleotide frequencies (*Brbić et al., 2015*), to avoid circularity these would need to be taken from intergenic sequences, with care needed to avoid influence from unannotated protein-coding genes or even pseudogenes.

Note that if a species were to experience a sudden reduction in census population size, for example due to habitat loss, leading to less effective selection, it would take some multiple of the neutral coalescent time for CAIS to fully adjust. CAIS thus represents a relatively long-term historical pattern of adaptation. The timescales setting neutral polymorphism-based $N_e$ estimates are likely shorter, based on a single round of coalescence. It is possible that the reason that we obtained correlations when we controlled for genome-wide GC content, but not when we controlled for local GC content, is also that codon adaptation adjusts slowly relative to the timescale of fluctuations in local GC content.

Here, we developed a new metric of species adaptedness at the codon level, capable of quantifying degrees of codon adaptation even among vertebrates. We chose vertebrates partly due to the abundance of suitable data, and partly as a stringent test case, given past studies finding limited evidence for codon adaptation (*Kessler and Dean, 2014*). It remains to be seen how CAIS behaves among species with stronger codon adaptation. We restricted our analysis to only the best annotated genomes, in part to ensure the quality of intergenic %GC estimates, and in part limited by the feasibility of running linear mixed models with 6 million datapoints. The phylogenetic tree is well resolved for vertebrate species, with an overrepresentation of mammalian species. Despite the focus on vertebrates, we were able to discover new results regarding selection on ISD.

Our finding that more effective selection prefers higher ISD was unexpected, given that lower-$N_e$ eukaryotes have more disordered proteins than higher-$N_e$ prokaryotes (*Ahrens et al., 2017*; *Basile et al., 2019*). However, this can be reconciled in a model in which highly disordered sequences are less likely to be found in high-$N_e$ species, but the sequences that are present tend to have slightly higher disorder than their low-$N_e$ homologs. High ISD might help mitigate the trade-off between affinity and specificity in protein–protein interactions (*Dunker et al., 1998*; *Huang and Liu, 2013*; *Lazar et al., 2022*); non-specific interactions might be short-lived due to the high entropy associated with disorder, which specific interactions are robust to.

Codon adaptation metrics more directly quantify how species vary in their exquisiteness of adaptation, than do estimates of effective population size that are based on neutral polymorphism. Both CAIS and ENC can also be estimated for far more species because they do not require polymorphism or mutation rate data, nor tRNA gene copy numbers and abundances, but only a single complete genome. CAIS has the additional advantage of not being confounded with amino acid frequencies. This makes CAIS a useful tool for applying nearly neutral theory to protein evolution, as shown by our worked example of ISD.

## Methods

**Key resources table**

| Reagent type (species) or resource | Designation | Source or reference | Identifiers | Additional information |
|---|---|---|---|---|
| Software, algorithm | IUPRED2 | DOI: https://doi.org/10.1093/nar/gky384 | RRID:SCR_014632 | |
| Software, algorithm | Codon Adaptation Index of Species | This paper | | See Materials and methods |
| Software, algorithm | Codon Adaptation Index | DOI: https://doi.org/10.1093/nar/15.3.1281 | | |
| Software, algorithm | ape | DOI: https://doi.org/10.1093/bioinformatics/bty633 | RRID:SCR_017343 | R package |
| Software, algorithm | Effective Number of Codons | DOI: https://doi.org/10.1093/oxfordjournals.molbev.a004201 | | |

### Species

Pfam sequences and IUPRED2 estimates of ISD predictions were taken from *James et al., 2021*, who studied species marked as 'Complete' in the GOLD database, with divergence dates available in Time-Tree (*Kumar et al., 2017*). *James et al., 2021* applied a variety of quality controls to exclude contaminants from the set of Pfams and assign accurate dates of Pfam emergence. Pfams that emerged prior to LECA are classified here as 'old', and Pfams that emerged after the divergence of animals and fungi from plants are classified as 'young', following annotation by *James et al., 2021*. Species list and other information can be found at https://github.com/MaselLab/Codon-Adaptation-Index-of-Species (copy archived at *MaselLab, 2024*).

### Codon Adaptation Index

*Sharp and Li, 1987* quantified codon bias through the CAI, a normalized geometric mean of synonymous codon usage bias across sites, excluding stop and start codons. We modify this to calculate CAI including stop and start codons, because of documented preferences among stop codons in mammals (*Wangen and Green, 2020*). While usually used to compare genes within a species, among-species comparisons can be made using a reference set of genes that are highly expressed (*Sharp and Li, 1987*). Each codon *i* is assigned an RSCU value:

$$RSCU_i = \frac{N_i}{\frac{1}{n_a} \sum_{j=1}^{n_a} N_j}, \tag{1}$$

where $N_i$ denotes the number of times that codon $i$ is used, and the denominator sums over all $n_a$ codons that code for that specific amino acid. RSCU values are normalized to produce a relative adaptiveness values $w_i$ for each codon, relative to the best adapted codon for that amino acid:

$$w_i \equiv \frac{RSCU_i}{RSCU_{max}}. \tag{2}$$

Let $L$ be the number of codons across all protein-coding sequences considered. Then

$$CAI = \left[ \Pi_{i=1}^{L} w_i \right]^{\frac{1}{L}}. \tag{3}$$

To understand the effects of normalization, it is useful to rewrite this as:

$$CAI = \left[ \Pi_{i=1}^{L} \frac{RSCU_i}{RSCU_{max}} \right]^{\frac{1}{L}} = \frac{CAI_{raw}}{CAI_{max}}, \tag{4}$$

where $CAI_{raw}$ is the geometric mean of the 'unnormalized' or observed synonymous codon usages, and $CAI_{max}$ is the maximum possible CAI given the observed codon frequencies.

## GC content

We calculated total %GC content (intergenic and genic) during a scan of all six reading frames across genic and intergenic sequences available from NCBI with access dates between May and July 2019 (described in *James et al., 2023*). Of the 170 vertebrates meeting the quality criteria of *James et al., 2021*, 118 had annotated intergenic sequences within NCBI, so we restricted the dataset further to keep only the 118 species for which total GC content was available.

## Codon Adaptation Index of Species

### Controlling for GC bias in synonymous codon usage

Consider a sequence region $r$ within species $s$ where each nucleotide has an expected probability of being G or C = $g_r$. For our main analysis, we consider just one region $r$ encompassing the entire genome of a species $s$. In a secondary analysis, we break the genome up and use local values of $g_r$ in the non-coding regions within and surrounding a gene or set of overlapping genes. To annotate the boundaries of these local regions, we first selected 1500 base pairs flanking each side of every coding sequence identified by NCBI annotations. Coding sequence annotations are broken up according to exon by NCBI. When coding sequences of the same gene did not fall within 3000 base pairs of each other, they were treated as different regions. When two coding sequences, whether from the same gene or from different genes, had overlapping 1500 bp catchment areas, we merged them together. $g_r$ was then calculated based on the non-coding sites within each region, including both genic regions such as promoters and non-genic regions such as introns and intergenic sequences.

With no bias between C vs. G, nor between A vs. T, nor patterns beyond the overall composition taken one nucleotide at a time, the expected probability of seeing codon $i$ in a triplet within $r$ is

$$p_{i,r} = \frac{g_r}{2}^{k_{GC}} \left( 1 - \frac{g_r}{2} \right)^{k_{AT}}, \tag{5}$$

where $k_{GC} + k_{AT} = 3$ total positions in codon $i$. The expected probability that amino acid $a$ in region $r$ is encoded by codon $i$ is

$$E_{i,r} = \frac{p_{i,r}}{\sum_{j=1}^{n_a} p_{j,r}}. \tag{6}$$

We can then measure the degree to which the observed codon frequencies diverge from these expected probabilities using the Kullback–Leibler divergence. This gives a CAIS metric for a species $s$ where $O_{i,s}$ is the observed frequency of codon $i$:

$$CAIS(s) = \Sigma_{i=1}^{64} O_{i,s} \log \left( \frac{O_{i,s}}{E_{i,s}} \right). \tag{7}$$

### Controlling for amino acid composition

Some amino acids may be more intrinsically prone to codon bias. We want a metric that quantifies effectiveness of selection (not amino acid frequency), so we re-weight CAIS on the basis of a standardized amino acid composition, to remove the effect of variation among species in amino acid frequencies.

Let $F_a$ be the frequency of amino acid $a$ across the entire dataset of 118 vertebrate genomes. We want to re-weight $O_{i,s}$ on the basis of $F_a$ to ensure that differences in amino acid frequencies among species do not affect CAIS, while preserving relative codon frequencies for the same amino acid. We do this by solving for $\alpha_{a,s}$ so that

$$F_a = \alpha_{a,s} \Sigma_{j=1}^{n_a} O_{j,s}. \tag{8}$$

We then define $f'_{i,s} = \alpha_{a,s} O_{i,s}$ to obtain an amino acid frequency adjusted CAIS:

$$CAIS\left(S\right) = \Sigma_{i=1}^{64} f'_{i,s}\, \log\left(\frac{O_{i,s}}{E_{i,s}}\right). \tag{9}$$

The $F_a$ values for our species set are at https://github.com/MaselLab/Codon-Adaptation-Index-of-Species/blob/main/CAIS_ENC_calculation/Total_amino_acid_frequency_vertebrates.txt. Use of the standardized set of amino acid frequencies $F_a$ has only a small effect on computed CAIS values relative to using each vertebrate species' own amino acid frequencies (*Figure 3—figure supplement 3*).

CAIS corrected for local intergenic GC content but not species-wide amino acid composition is

$$CAIS_{localGC}\left(s\right) = \left(\Pi_{r=1}^{G}\Sigma_{i=1}^{64} O_{i,r}\, \log\left(\frac{O_{i,r}}{E_{i,r}}\right)\right)^{\frac{1}{L}}, \tag{10}$$

where $O_{i,r}$ is the number of times codon $i$ appears in region $r$ of species $s$, $E_{i,r}$ is the expected number of times codon $i$ would appear in region $r$ of species $s$ given the local intergenic GC content, $G$ is the number of regions, and $L = \sum_{r=1}^{G}\sum_{i=1}^{64} O_{i,r}$ is the total number of codons in the genome. Rewritten for greater computational ease:

$$CAIS_{localGC}\left(s\right) = e^{\frac{1}{L}\Sigma_{r=1}^{G}\ln\left(\sum_{i=1}^{64} O_{i,r}\, \log\left(\frac{O_{i,r}}{E_{i,r}}\right)\right)}. \tag{11}$$

Given the limited impact of amino acid frequency correction, we used *Equation 11* for the local GC results, but we could correct for amino acid composition by replacing the $O_{i,r}$ prefactor with $f'_{i,s}$, or even $f'_{i,r}$.

## Novembre's ENC controlled for total GC content

The expected number of codons is based on the squared deviations $X_a^2$ of the frequencies of the codons for each amino acid $a$ from null expectations:

$$X_a^2 = \Sigma_{i=1}^{n_a}\frac{N_a\left(O_i - E_i\right)^2}{E_i}, \tag{12}$$

where $N_a$ is the total number of times that amino acid $a$ appears. *Novembre, 2002* defines the corrected '$F$ value' of amino acid $a$ as

$$\widehat{F'}_a = \frac{X_a^2 + N_a - n_a}{n_a\left(N_a - 1\right)} \tag{13}$$

and

$$ENC = 2 + \frac{9}{\widehat{F'}_2} + \frac{1}{\widehat{F'}_3} + \frac{5}{\widehat{F'}_4} + \frac{3}{\widehat{F'}_6}, \tag{14}$$

where each $F'_{n_a}$ is the average of the '$F$ values' for amino acids with $n_a$ synonymous codons. Past measures of ENC do not contain stop or start codons (*Wright, 1990*; *Novembre, 2002*; *Fuglsang, 2004*), but as we did for CAI and CAIS above, we include stop codons as an 'amino acid' and therefore amend *Equation 14* to

$$ENC = 2 + \frac{9}{\widehat{F'}_2} + \frac{2}{\widehat{F'}_3} + \frac{5}{\widehat{F'}_4} + \frac{3}{\widehat{F'}_6}. \tag{15}$$

## Statistical analysis

All statistical modeling was done in R 3.5.1. Scripts for calculating CAI and CAIS were written in Python 3.7.

## Phylogenetic Independent Contrasts

Spurious phylogenetically confounded correlations can occur when closely related species share similar values of both metrics. One danger of such pseudoreplication is Simpson's paradox, where there are negative slopes within taxonomic groups, but a positive slope among them might combine to yield an overall positive slope. We avoid pseudoreplication by using Phylogenetic Independent Contrasts (PIC) (*Felsenstein, 1985*) to assess correlation. PIC analysis was done using the R package 'ape' (*Paradis and Schliep, 2019*).

## Acknowledgements

We thank Luke Kosinski, David Liberles, and Sawsan Wehbi for helpful discussions, Paul Nelson for providing the genome-wide GC contents, and the University of Arizona Undergraduate Biology Research Program for training. We thank Gavin Douglas for writing a convenient end-to-end implementation of CAIS on the basis of our preprint, which can be found at https://github.com/gavin-mdouglas/handy_pop_gen/blob/main/CAIS.py, and for catching a minor bug in our code in time for us to correct it in the version of record. We thank the anonymous reviewers for constructive feedback, and Laurent Duret for helpful elaboration on the concerns of reviewer 1.

## Additional information

### Funding

| Funder | Grant reference number | Author |
|---|---|---|
| National Institutes of Health | GM104040 | Catherine A Weibel<br>Jennifer E James<br>Sara M Willis<br>Joanna Masel |
| National Institutes of Health | GM132008 | Andrew L Wheeler |
| John Templeton Foundation | 60814 | Catherine A Weibel<br>Jennifer E James<br>Sara M Willis<br>Joanna Masel |
| Arnold and Mabel Beckman Foundation | Scholars Program | Catherine A Weibel |
| National Science Foundation | WAESO/LSAMP Cooperative Agreement HRD-1101728 | Catherine A Weibel |
| National Aeronautics and Space Administration | Arizona NASA Space Grant Consortium, Cooperative Agreement 80NSSC20M0041 | Catherine A Weibel |
| National Science Foundation | Graduate Research Fellowship Program | Hanon McShea |

The funders had no role in study design, data collection, and interpretation, or the decision to submit the work for publication.

### Author contributions

Catherine A Weibel, Conceptualization, Data curation, Formal analysis, Funding acquisition, Investigation, Visualization, Methodology, Writing – original draft, Writing – review and editing; Andrew L Wheeler, Formal analysis, Investigation, Visualization, Methodology, Writing – review and editing; Jennifer E James, Resources, Data curation, Supervision, Investigation, Methodology, Writing – original draft; Sara M Willis, Resources, Data curation, Supervision; Hanon McShea, Methodology, Writing – review and editing; Joanna Masel, Conceptualization, Formal analysis, Supervision, Funding acquisition, Methodology, Writing – original draft, Project administration, Writing – review and editing

Author ORCIDs

Catherine A Weibel (iD) http://orcid.org/0000-0003-1837-5209
Andrew L Wheeler (iD) http://orcid.org/0000-0002-5347-5419
Jennifer E James (iD) http://orcid.org/0000-0003-0518-6783
Sara M Willis (iD) https://orcid.org/0000-0002-1605-6426
Hanon McShea (iD) http://orcid.org/0000-0002-9341-4899
Joanna Masel (iD) https://orcid.org/0000-0002-7398-2127

Reviewer #2 (Public Review): https://doi.org/10.7554/eLife.87335.3.sa1
Author response https://doi.org/10.7554/eLife.87335.3.sa2

---

## Additional files

### Supplementary files
• MDAR checklist

### Data availability

There is no new data. Processed data and code underlying this article are available in the public repository at https://github.com/MaselLab/Codon-Adaptation-Index-of-Species (copy archived at *MaselLab, 2024*).

The following previously published dataset was used:

| Author(s) | Year | Dataset title | Dataset URL | Database and Identifier |
|---|---|---|---|---|
| Jennifer J, Sara W, Paul N, Catherine W, Luke K, Joanna M | 2020 | Data from: Universal and taxon-specific trends in protein sequences as a function of age | https://doi.org/10.6084/m9.figshare.12037281.v1 | figshare, 10.6084/m9.figshare.12037281.v1 |

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
