## [Editor Report · eLife assessment]

This study develops a **useful** metric for quantifying codon usage adaptation - the Codon Adaptation Index of Species (CAIS). This metric permits direct comparisons of the strength of selection at the molecular level across species. The study is based on **solid** evidence, and the authors identify relationships between CAIS and the presence of disordered protein domains. Other correlations, such as the one between CAIS and body size, are weak and non-significant. In summary, the study introduces an interesting new approach to quantifying codon usage across species, which may be helpful in attempts to measure selection at the molecular level.

---

## [Referee Report · Reviewer #2 (Public Review)]

Summary:

The goal of the authors in this study is to develop a more reliable approach for quantifying codon usage such that it is more comparable across species. Specifically, the authors wish to estimate the degree of adaptive codon usage, which is potentially a general proxy for the strength of selection at the molecular level. To this end, the authors created the Codon Adaptation Index for Species (CAIS) that attempts to control for differences in amino acid usage and GC% across species. Using their new metric, the authors observe a positive relationship between CAIS and the overall “disorderedness” of a species protein domains. I think CAIS has the potential to be a valuable tool for those interested in comparing codon adaptation across species in certain situations. However, I have certain theoretical concerns about CAIS as a direct proxy for the efficiency of selection sNe when mutation bias changes across species.

Strengths:

(1) I appreciate that the authors recognize the potential issues of comparing CAI when amino acid usage varies and correct for this in CAIS. I think this is sometimes an under-appreciated point in the codon usage literature, as CAI is a relative measure of codon usage bias (i.e. only considers synonyms). However, the strength of natural selection on codon usage can potentially vary across amino acids, such that comparing mean CAI between protein regions with different amino acid biases may result in spurious signals of statistical significance.

(2) The CAIS metric presented here is generally applicable to any species that has an annotated genome with protein-coding sequences. A significant improvement over the previous version is the implementation of software tool for applying this method.

(3) The authors do a better job of putting their results in the context of the underlying theory of CAIS compared to the previous version.

(4) The paper is generally well-written.

Weaknesses:

(1) The previously observed correlation between CAIS and body size was due to a bug when calculating phylogenetic independent contrasts. I commend the authors for acknowledging this mistake and updating the manuscript accordingly. I feel that the unobserved correlation between CAIS and body size should remain in the final version of the manuscript. Although it is disappointing that it is not statistically significant, the corrected results are consistent with previous findings (Kessler and Dean 2014).

(2) I appreciate the authors for providing a more detailed explanation of the theoretical basis model. However, I remain skeptical that shifts in CAIS across species indicates shifts in the strength of selection. I am leaving the math from my previous review here for completeness.

As in my previous review, let’s take a closer look at the ratio of observed codon frequencies vs. expected codon frequencies under mutation alone, which was previously notated as RSCUS in the original formulation. In this review, I will keep using the RSCUS notation, even though it has been dropped from the updated version. The key point is this is the ratio of observed and expected codon frequencies. If this ratio is 1 for all codons, then CAIS would be 0 based on equation 7 in the manuscript – consistent with the complete absence of selection on codon usage. From here on out, subscripts will only be used to denote the codon and it will be assumed that we are only considering the case of r = genome for some species s.RSCUSi=OiEi

I think what the authors are attempting to do is “divide out” the effects of mutation bias (as given by Ei), such that only the effects of natural selection remain, i.e. deviations from the expected frequency based on mutation bias alone represents adaptive codon usage. Consider Gilchrist et al. GBE 2015, which says that the expected frequency of codon i at selection-mutation-drift equilibrium in gene g for an amino acid with Na synonymous codons isEi,g=e−ΔMi−Δηiϕg∑j=1Nae−ΔMj−Δηjϕg

where ∆M is the mutation bias, ∆η is the strength of selection scaled by the strength of drift, and φg is the gene expression level of gene g. In this case, ∆M and ∆η reflect the strength and direction of mutation bias and natural selection relative to a reference codon, for which ∆M,∆η = 0. Assuming the selection-mutation-drift equilibrium model is generally adequate to model of the true codon usage patterns in a genome (as I do and I think the authors do, too), the Ei,g could be considered the expected observed frequency codon i in gene g

E[Oi,g].

Let’s re-write the Ei=pi∑j=1Napj in the form of Gilchrist et al., such that it is a function of mutation bias ∆M. For simplicity we will consider just the two codon case and assume the amino acid sequence is fixed. Assuming GC% is at equilibrium, the term gr and 1 − gr can be written asgr=μAT→GCμAT→GC+μGC→AT1−gr=μGC→ATμAT→GC+μGC→AT

where µx→y is the mutation rate from nucleotides x to y. As described in Gilchrist et al. MBE 2015 and

Shah and Gilchrist PNAS 2011, the mutation bias ΔMNNA,NNG=log⁡(μAT→GCμGC→AT) .This can be expressed in terms of the equilibrium GC content by recognizing thatgr1−gr=μAT→GCμGC→AT⟹gr1−gr=eΔM

As we are assuming the amino acid sequence is fixed, the probability of observing a synonymous codon i at an amino acid becomes just a Bernoulli process.pi=grx(1−gr)(1−x)

If we do this, thenENNA=pNNApNNA+pNNG=1−grgr+(1−gr)=1gr1−gr+1=1eΔM+1=e−ΔM1+e−ΔM

Recall that in the Gilchrist et al. framework, the reference codon has ∆MNNG,NNG = 0 = ⇒ e−∆MNNG,NNG =

(1) Thus, we have recovered the Gilchrist et al. model from the formulation of Ei under the assumption that natural selection has no impact on codon usage and codon NNG is the pre-defined reference codon. To see this, plug in 0 for ∆η in equation (1).

We can then calculate the expected RSCUS using equation (1) (using notation E[Oi]) and equation (6) for the two codon case. For simplicity assume, we are only considering a gene of average expression defined as (ϕg=1). Assume in this case that NNG is the reference codon (∆MNNG,∆ηNNG = 0).E[RSCUSNNA]=E[ONNA]ENNA=e−ΔηNNA(e−ΔMNNA+e−ΔMNNG)e−ΔMNNA−ΔηNNA+e−ΔMNNG−ΔηNNG=e−ΔMNNA−ΔηNNA+e−ΔMNNG−ΔηNNAe−ΔMNNA−ΔηNNA+e−ΔMNNG−ΔηNNG=e−ΔMNNA−ΔηNNA+e−ΔηNNAe−ΔMNNA−ΔηNNA+1

This shows that the expected value of RSCUS for a two codon amino acid is expected to increase as the strength of selection ∆η increases, which is desired. Note that ∆η in Gilchrist et al. is formulated in terms of selection against a codon relative to the reference, such that a negative value represents that a codon is favored relative to the reference. If ∆η = 0 (i.e. selection does not favor either codon), then E[RSCUS] = 1. Also note that the expected RSCUS does not remain independent of the mutation bias. This means that even if sNe (i.e. the strength of natural selection) does not change between species, changes to the strength and direction of mutation bias across species could impact RSCUS. Assuming my math is right, I think one needs to be cautious when interpreting CAIS as representative of the differences in the efficiency of selection across species except under very particular circumstances.

Consider our 2-codon amino acid scenario. You can see how changing GC content without changing selection can alter the CAIS values calculated from these two codons. Particularly problematic appears to be cases of extreme mutation biases, where CAIS tends toward 0 even for higher absolute values of the selection parameter. Codon usage for the majority of the genome will be primarily determined by mutation biases,

with selection being generally strongest in a relatively few highly-expressed genes. Strong enough mutation biases ultimately can overwhelm selection, even in highly-expressed genes, reducing the fraction of sites subject to codon adaptation.

**Review image 1. sa1fig1:** 

**Review image 2. sa1fig2:** CAIS (Low Expression).

**Review image 3. sa1fig3:** CAIS (Average Expression).

**Review image 4. sa1fig4:** CAIS (High Expression).

If we treat the expected codon frequencies as genome-wide frequencies, then we are basically assuming this genome made up entirely of a single 2-codon amino acid with selection on codon usage being uniform across all genes. This is obviously not true, but I think it shows some of the potential limitations of the CAIS approach. Based on these simulations, CAIS seems best employed under specific scenarios. One such case could be when it is known that mutation bias varies little across the species of interest. Looking at the species used in this manuscript, most of them have a GC content around 0.41, so I suspect their results are okay (assuming things like GC-biased gene conversion are not an issue). Outliers in GC content probably are best excluded from the analysis.

Although I have not done so, I am sure this could be extended to the 4 and 6 codon amino acids. One potential challenge to CAIS is the non-monotonic changes in codon frequencies observed in some species (again, see Shah and Gilchrist 2011 and Gilchrist et al. 2015).

---

## [Author Response]

The following is the authors’ response to the original reviews.

In addition to our responses to reviewer suggestions below, a minor bug in the calculation of CAIS was brought to our attention by a reader of our preprint. We have corrected this bug and rerun analyses, whose results became slightly stronger as noise was removed. While we were doing that, someone pointed out to us that our equations were almost the same as Kullback-Leibler divergence, which explains why our metric performed so well. We have made the numerically trivial (see before vs. after figure below) mathematical change to use Kullback-Leibler divergence instead, and now have a better story, with a solid basis in information theory, as to why CAIS works.

**Author response image 1. sa2fig1:** 

Unfortunately, we discovered a second bug that caused our PIC correction code to fail to perform the needed correction for phylogenetic confounding. The previously reported correlation between CAIS (or ENC) with body mass no longer survives PIC-correction. We have therefore removed this analysis from the manuscript. Our story now stands more on the theoretical basis of CAIS and ENC than on the post facto validation than it previously did. We now also present CAIS and ENC on a more equal footing. ENC results are slightly stronger, while CAIS has the complementary advantage of correcting for amino acid frequencies.

The work involved in these changes, as well as some of the responses to reviews below, justifies changing the second author into a co-first author, and adding an additional coauthor (Hanon McShea) who discovered the second bug.

**Reviewer #1 (Public Review):**
In this manuscript, the authors propose a new codon adaptation metric, Codon Adaptation Index of Species (CAIS), which they present as an easily obtainable proxy for effective population size. To permit between-species comparisons, they control for both amino acid frequencies and genomic GC content, which distinguishes their approach from existing ones. Having confirmed that CAIS negatively correlates with vertebrate body mass, as would be expected if small-bodied species with larger effective populations experience more efficient selection on codon usage, they then examine the relationship between CAIS and intrinsic structural disorder in proteins.The idea of a robust species-level measure of codon adaptation is interesting. If CAIS is indeed a reliable proxy for the effectiveness of selection, it could be useful to analyze species without reliable life history- or mutation rate data (which will apply to many of the genomes becoming available in the near future).A key question is whether CAIS, in fact, measures adaptation at the codon level. Unfortunately, CAIS is only validated indirectly by confirming a negative correlation with body mass. As a result, the observations about structural disorder are difficult to evaluate.

As discussed in the preamble above, we have replaced the body mass validation with a stronger theoretical basis in information theory.

A potential problem is that differences in GC between species are not independent of life history. Effective population size can drive compositional differences due to the effects of GC-biased gene conversion (gBGC). As noted by Galtier et al. (2018), genomic GC correlates negatively with body mass in mammals and birds. It would therefore be important to examine how gBGC might affect CAIS, and to what extent it could explain the relationship between CAIS and body mass.Suppose that gBGC drives an increase in GC that is most pronounced at 3rd codon positions in highrecombination regions in small-bodied species. In this case, could observed codon usage depart more strongly from expectations calculated from overall genomic GC in small vertebrates compared to large ones? The authors also report that correcting for local intergenic GC was unsuccessful, based on the lack of a significant negative relationship with body mass (Figure 3D). In principle, this could also be consistent with local GC providing a relatively more appropriate baseline in regions with high recombination rates. Considering these scenarios would clarify what exactly CAIS is capturing.

Figure 3 (previously Supplementary Figures S5A and S5B) shows that CAIS is negligibly correlated with %GC (not robust to multiple comparisons correction), and ENC not at all. We believe this is evidence against the possibility brought up by the reviewer, i.e. that Ne might affect gBGC (and hence global %GC). This relationship, if present, could act as a confounding effect, but it is not present within our species dataset.

Note that we expect our genomic-GC-based codon usage expectations to reflect unchecked gBGC in an average genomic region, independently of whether that species has high or low Ne. Our working model is that non-selective forces, include gBGC as well as conventional mutation biases, vary among species, and that they rather than selection determine each species’ genome-wide %GC. By correcting for genome-wide %GC, CAIS and ENC correct for both mutation bias and gBGC, in order to isolate the effects of selection.

This argument, based on an average genomic region, is vulnerable to gene-rich genomic regions having differentially higher recombination rates and hence GC-biased gene conversion. However, we do not see the expected positive correlation between |𝐥𝐨𝐜𝐚𝐥 𝐆𝐂 - global GC| and CAIS (see new Figure 5), again suggesting that gene conversion strength is not a confounding factor acting on CAIS.

Given claims about "exquisitely adapted species", the case for using CAIS as a measure of codon adaptation would also be stronger if a relationship with gene expression could be demonstrated. RSCU is expected to be higher in highly expressed genes. Is there any evidence that the equivalent GCcontrolled measure behaves similarly?

Correlations with gene expression are outside the scope of the current work, which is focused on producing and exploiting a single value of codon adaptation per species. It is indeed possible that our general approach of using Kullback-Leibler divergence to correct for genomic %GC could be useful in future work investigating differences among genes.

The manuscript is overall easy to follow, though some additional context may be helpful for the general reader. A more detailed discussion of how this work compares to the approach taken by Galtier et al. (2018), which accounted for GC content and gBGC when examining codon preferences, would be appropriate, for example. In addition, it would have been useful to mention past work that has attempted to explicitly quantify selection on codon usage.

One key difference between our work and that of Galtier et al. 2018 is that our approach does not rely on identifying specific codon preferences as a function of species. Our approach might therefore be robust to scenarios where different genes have different codon preferences (see Gingold et al. 2014 https://doi.org/10.1016/j.cell.2014.08.011). At a high level, our results are in broad agreement with those of Galtier et al., 2018, who found that gBGC affected all animal species, regardless of Ne, and who like us, found that the degree of selection on codon usage depended on Ne.

**Reviewer #2 (Public Review):**
## SummaryThe goal of the authors in this study is to develop a more reliable approach for quantifying codon usage such that it is more comparable across species. Specifically, the authors wish to estimate the degree of adaptive codon usage, which is potentially a general proxy for the strength of selection at the molecular level. To this end, the authors created the Codon Adaptation Index for Species (CAIS) that controls for differences in amino acid usage and GC% across species. Using their new metric, the authors find a previously unobserved negative correlation between the overall adaptiveness of codon usage and body size across 118 vertebrates. As body size is negatively correlated with effective population size and thus the general strength of natural selection, the negative correlation between CAIS and body size is expected. The authors argue this was previously unobserved due to failures of other popular metrics such as Codon Adaptation Index (CAI) and the Effective Number of Codons (ENC) to adequately control for differences in amino acid usage and GC content across species. Most surprisingly, the authors also find a positive relationship between CAIS and the overall "disorderedness" of a species protein domains. As some of these results are unexpected, which is acknowledged by the authors, I think it would be particularly beneficial to work with some simulated datasets. I think CAIS has the potential to be a valuable tool for those interested in comparing codon adaptation across species in certain situations. However, I have certain theoretical concerns about CAIS as a direct proxy for the efficiency of selection Ne when the mutation bias changes across species.## Strengths(1) I appreciate that the authors recognize the potential issues of comparing CAI when amino acid usage varies and correct for this in CAIS. I think this is sometimes an under-appreciated point in the codon usage literature, as CAI is a relative measure of codon usage bias (i.e. only considers synonyms). However, the strength of natural selection on codon usage can potentially vary across amino acids, such that comparing mean CAI between protein regions with different amino acid biases may result in spurious signals of statistical significance (see Cope et al. Biochemica et Biophysica Acta - Biomembranes 2018 for a clear example of this).

We now cite Cope et al. as an example of how amino acid composition can act as a confounding factor.

(2) The authors present numerous analysis using both ENC and mean CAI as a comparison to CAIS, helping given a sense of how CAIS corrects for some of the issues with these other metrics. I also enjoyed that they examined the previously unobserved relationship between codon usage bias and body size, which has bugged me ever since I saw Kessler and Dean 2014. The result comparing protein disorder to CAIS was particularly interesting and unexpected.

Unfortunately, our previous PIC correction code was buggy, and in fact the relationship with body size does not survive PIC correction (although it is strong prior to PIC correction). We have therefore removed it from the paper. However, the more novel result on protein disorder remains strong.

(3) The CAIS metric presented here is generally applicable to any species that has an annotated genome with protein-coding sequences.## Weaknesses(1) The main weakness of this work is that it lacks simulated data to confirm that it works as expected. This would be particularly useful for assessing the relationship between CAIS and the overall effect of protein structure disorder, which the authors acknowledge is an unexpected result. I think simulations could also allow the authors to assess how their metric performs in situations where mutation bias and natural selection act in the same direction vs. opposite directions. Additionally, although I appreciate their comparisons to ENC and mean CAI, the lack of comparison to other popular codon metrics for calculating the overall adaptiveness of a genome (e.g. dos Reis et al.'s S statistic, which is a function of tRNA Adaptation Index (tAI) and ENC) may be more appropriate. Even if results are similar to S, CAIS has a noted advantage that it doesn't require identifying tRNA gene copy numbers or abundances, which I think are generally less readily available than genomic GC% and protein-coding sequences.

The main limitation of dos Reis’s test in our view is that, like the better versions of CAI, it requires comparable orthologs across species. See also the discussion below re the benefits of proteome-wide approach. We now also note the advantage of not needing tRNA gene copy numbers and abundances.

Simulated datasets would be great, but we think it a nice addition rather than must-have, in particular because we are skeptical about whether our understanding of all relevant processes is good enough such that simulations would add much to our more heuristic argument along the lines of Figure 2. E.g. the complications of Gingold et al. 2014 cited above are pertinent, but incorporating them would make simulations quite involved. Instead, we now have a stronger theoretical justification for CAIS grounded in information theory. We have significantly expanded discussion of Figure 2 to give a clearer idea of the conceptual underpinnings of CAIS and ENC.

The authors mention the selection-mutation-drift equilibrium model, which underlies the basic ideas of this work (e.g. higher Ne results in stronger selection on codon usage), but a more in-depth framing of CAIS in terms of this model is not given. I think this could be valuable, particularly in addressing the question "are we really estimating what we think we're estimating?"Let's take a closer look at the formulation for RSCUS. From here on out, subscripts will only be used to denote the codon and it will be assumed that we are only considering the case of r = genome for some species s.RSCUSi=OiEiI think what the authors are attempting to do is "divide out" the effects of mutation bias (as given by Ei), such that only the effects of natural selection remain, i.e. deviations from the expected frequency based on mutation bias alone represent adaptive codon usage. Consider Gilchrist et al. MBE 2015, which says that the expected frequency of codon i at selection-mutation-drift equilibrium in gene g for an amino acid with Na synonymous codons isEi,g=e−ΔMi−Δηiϕg∑j=1Nae−ΔMjΔηjϕgwhere ∆M is the mutation bias, ∆η is the strength of selection scaled by the strength of drift, and φg is the gene expression level of gene g. In this case, ∆M and ∆η reflect the strength and direction of mutation bias and natural selection relative to a reference codon, for which ∆M,∆η = 0. Assuming the selection-mutation-drift equilibrium model is generally adequate to model of the true codon usage patterns in a genome (as I do and I think the authors do, too), the Ei,g could be considered the expected observed frequency codon i in gene g E[Oi,g].Let’s re-write the Ei=pi∑j=1Napj in the form of Gilchrist et al., such that it is a function of mutation bias ∆M. For simplicity we will consider just the two codon case and assume the amino acid sequence is fixed. Assuming GC% is at equilibrium, the term gr and 1 − gr can be written asgr=μAT→GCμAT→GC+μGC→AT1−gr=μGC→ATμAT→GC+μGC→ATwhere µx→y is the mutation rate from nucleotides x to y. As described in Gilchrist et al. MBE 2015 and Shah and Gilchrist PNAS 2011, the mutation bias ΔMNNA,NNG=log(μAT→GCμGC→AT) .This can be expressed in terms of the equilibrium GC content by recognizing thatgr1−gr=μAT→GCμGC→ATimpliesgr1−gr=eΔMAs we are assuming the amino acid sequence is fixed, the probability of observing a synonymous codon i at an amino acid becomes just a Bernoulli process.pi=grx(1−gr)(1−x)If we do this, thenENNA=pNNApNNA+pNNG =1−grgr+(1−gr) =1gr1−gr+1 =1eΔM+1 =e−ΔM1+e−ΔMRecall that in the Gilchrist et al. framework, the reference codon has ∆MNNG,NNG = 0 = ⇒ e−∆MNNG,NNG = 1. Thus, we have recovered the Gilchrist et al. model from the formulation of Ei under the assumption that natural selection has no impact on codon usage and codon NNG is the pre-defined reference codon. To see this, plug in 0 for ∆η in equation (1)..We can then calculate the expected RSCUS using equation (1) (using notation E[Oi]) and equation (6) for the two codon case. For simplicity assume, we are only considering a gene of average expression (defined as ϕg=1). Assume in this case that NNG is the reference codon (∆MNNG,∆ηNNG = 0).E[RSCUSNNA]=E[ONNA]ENNA=e−ΔηNNA(e−ΔMNNA+e−ΔMNNG)e−ΔMNNA−ΔηNNA+e−ΔMNNGΔηNNG=e−ΔMNNA−ΔηNNA+e−ΔMNNG−ΔηNNAe−ΔMNNA−ΔηNNA+e−ΔMNNG−ΔηNNG=e−ΔMNNA−ΔηNNA+e−ΔηNNAeΔMNNA−ΔηNNA+1This shows that the expected value of RSCUS for a two-codon amino acid is expected to increase as the strength of selection Δη increases, which is desired. Note that Δη in Gilchrist et al. is formulated in terms of selection *against* a codon relative to the reference, such that a negative value represents that a codon is favored relative to the reference. If (i.e. selection does not favor either codon), then E[RSCUS]=1. Also note that the expected RSCUS does not remain independent of the mutation bias. This means that even if sNe (i.e. the strength of natural selection) does not change between species, changes to the strength and direction of mutation bias across species could impact RSCUS. Assuming my math is right, I think one needs to be cautious when interpreting CAIS as representative of the differences in the efficiency of selection across species except under very particular circumstances. One such case could be when it is known that mutation bias varies little across the species of interest. Looking at the species used in this manuscript, most of them have a GC content ranging around 0.41, so I suspect their results are okay.Although I have not done so, I am sure this could be extended to the 4 and 6 codon amino acids.

We thank Reviewer 2 for explicitly laying out the math that was implicit in our Figures 1 and 2. While we keep our more heuristic presentation, our revised manuscript now more clearly acknowledges that the per-site codon adaptation bias depicted in Figure 1 has limited sensitivity to s*Ne. The reason that we believe our approach worked despite this, is that we think the phenomenon is driven by what is shown in Figure 2. I.e., where Ne makes a difference is by determining the proteome-wide fraction of codons subject to significant codon adaptation, rather than by determining the strength of codon adaptation at any particular site or gene. We have made multiple changes to the texts to make this point clearer.

Another minor weakness of this work is that although the method is generally applicable to any species with an annotated genome and the code is publicly available, the code itself contains hard-coded values for GC% and amino acid frequencies across the 118 vertebrates. The lack of a more flexible tool may make it difficult for less computationally-experienced researchers to take advantage of this method.

Genome-wide %GC values are hard-coded because they were taken from the previous study of James et al. (2023) https://doi.org/10.1093/molbev/msad073. As summarized in the manuscript, genome-wide %GC was a byproduct of a scan of all six reading frames across genic and intergenic sequences available from NCBI with access dates between May and July 2019. The more complicated code used to calculate the intergenic %GC, and the code used to calculate amino acid frequencies is located at https://github.com/MaselLab/CodonAdaptation-Index-of-Species. Luckily, someone else just wrote a simpler end to end pipeline for us, on the basis of our preprint. We now note this in the Acknowledgements, and link to it: https://github.com/gavinmdouglas/handy_pop_gen/blob/main/CAIS.py.